# Visually Grounded Interpretation of Noun-Noun Compounds in English

**Inga Lang**[1], **Lonneke van der Plas**[1], **Malvina Nissim**[2] and **Albert Gatt**[3]

[1]Idiap Research Institute, `firstname.lastname@idiap.ch`
[2]University of Groningen, `m.nissim@rug.nl`
[3]Utrecht University, `a.gatt@uu.nl`

## Abstract

Noun-noun compounds (NNCs) occur frequently in the English language. Accurate NNC interpretation, i.e. determining the implicit relationship between the constituents of a NNC, is crucial for the advancement of many natural language processing tasks. Until now, computational NNC interpretation has been limited to approaches involving linguistic representations only. However, research suggests that grounding linguistic representations in vision or other modalities can increase performance on this and other tasks. Our work is a novel comparison of linguistic and visuo-linguistic representations for the task of NNC interpretation. We frame NNC interpretation as a relation classification task, evaluating on a large, relationally-annotated NNC dataset. We combine distributional word vectors with image vectors to investigate how visual information can help improve NNC interpretation systems. We find that adding visual vectors yields modest increases in performance on several configurations of our dataset. We view this as a promising first exploration of the benefits of using visually grounded representations for NNC interpretation.

## 1 Introduction

Conceptual combination is the cognitive process that allows us to combine two mental concepts into one, for example by juxtaposing or otherwise merging two concepts. For instance, a house located on a beach might typically be called a 'beach house'. Noun-noun compounds (NNCs) are the linguistic phenomenon in which two nouns are joined to form one single, syntactically inseparable unit. The process of combining nouns into new nominal units is both highly prevalent and infinitely productive in a language like English (Libben, 2014), and also exists in various forms in many other languages, including but not limited to German, Norwegian, Hindi, Tamil, Japanese, Chinese, Bulgarian, and

Turkish (Nakov, 2013). In English, the *head* of the NNC is usually the rightmost word, and determines the semantic category of the compound. The leftmost word in English NNCs is referred to as the *modifier*. Although NNCs are a common occurrence, the highly productive nature of compounding (Algeo and Algeo, 1993) means that individual NNCs tend to have relatively low frequency counts (Kim and Baldwin, 2006). Compositional models have therefore been of particular interest to researchers working on computational NNC representations (e.g. Shwartz, 2019; Dima, 2016).

Due to of the high prevalence and complex nature of English NNCs, the ability to interpret compounds would greatly improve several important natural language processing tasks, such as machine translation (Baldwin and Tanaka, 2004; Balyan and Chatterjee, 2015), text summarization (e.g. Silber and McCoy, 2000), question answering (e.g. Mann, 2002), and natural language inference (e.g. MacCartney and Manning, 2008).

In this paper, we frame compound interpretation as a classification problem. The goal is to identify the semantic relationship between the nominal elements of a compound. We explicitly compare the contribution of linguistic and multimodal (visuo-linguistic) representations to this task.[1] In part, the motivation for this is theoretical, as a computational account of linguistic meaning has to address the link between symbolic and non-symbolic information (Bender and Koller, 2020; Bisk et al., 2020). A further motivation is the empirical observation that grounding representations in vision gives rise to richer meaning representations (Bruni et al., 2012; Collell Talleda and Moens, 2016). Composition in the visual modality has also been shown to be possible for certain NNCs (Pezzelle et al., 2016). A final motivation comes from find-

---

[1]The code for our experiments, as well as our visual embeddings, can be found here: `https://github.com/ingalang/multimodal_NC_interpretation`

ings in cognitive science suggesting that visually grounded word representations yield results closer to human performance on some NNC processing tasks (Günther et al., 2020). Our goal is to assess to what extent visual grounding helps to accurately identify the semantic relationship between NNC constituents. For example, Figure 1 displays images of the constituents of 'beach house' as well as the compound itself. Does the relationship between the constituents in the NNC become easy to predict once such visual information is incorporated, in addition to the textual representation of the constituents?

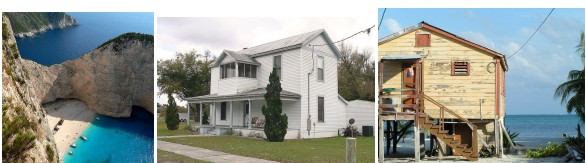

Figure 1: Picture of a beach, a house, and a beach house from ImageNet.

## 2   Background

Early approaches to the automatic interpretation of noun compounds included rule-based approaches (Finin, 1980; Vanderwende, 1994) or semi-automatic approaches requiring some user interaction (Barker and Szpakowicz, 1998). Other work utilized frequency statistics of NNC constituents to build probabilistic models for NNC interpretation (Lauer, 1995; Lapata and Keller, 2004). Kim and Baldwin (2005, 2006) leveraged WordNet (Miller, 1998) similarities in supervised training approaches.

Some approaches to NNC interpretation deal with identifying an appropriate paraphrase for a compound which explicitly states the relation between the compound's constituents. Several paraphrasing-based approaches have viewed the task of freely paraphrasing noun compounds as a goal in itself (Hendrickx et al., 2019; Ponkiya et al., 2020; Shwartz and Dagan, 2019; Van de Cruys et al., 2013), whereas others have used paraphrases as inputs to a model, representing NNCs in some way through their paraphrases.

Other approaches to NNC relation classification tend to be centered around classifying NNCs based on a pre-defined set of compound relations using various representations of the compounds themselves as input. Both compositional and distributional representations have been tested. Dima

(2016) and Shwartz (2019) both tested various ways of representing noun compounds. Dima (2016) performed the first experiments on compositional representations of English NNCs, using compositional models such as the FullAdd model (Zanzotto et al., 2010) and the Matrix model (Socher et al., 2012). Dima's results, which were tested on datasets by Tratz and Hovy (2010) and Ó Séaghdha (2008), reached a similar performance to the results obtained by the creators of said datasets, respectively. Yet, Dima's work utilized simpler methods and did not include lexical and relational information, as opposed to Tratz and Hovy and Ó Séaghdha.

Visuo-linguistic representations for NNC interpretation have received far less attention. Günther et al. (2020) created the first computational model of visuo-linguistic conceptual combination, reporting positive results on several NNC processing tasks. Pezzelle et al. (2016) found that certain compounds can be composed in the visual domain by simple addition of image feature vectors. However, none of these studies have touched upon NNC *interpretation* using visuo-linguistic data, an area that remains unexplored, to our knowledge.

The present work focuses on the interpretation of NNCs that possess at least some degree of compositionality. This is justified on the grounds that novel compounds, which are very common (Algeo and Algeo, 1993), must be interpreted compositionally on first encounter. We employ one compositional model, called the Full Additive model (Zanzotto et al., 2010), as well as simple vector concatenation, in our experiments to construct compound vectors from individual constituent vectors. We do this for linguistic and visual vectors separately, and then combine the two modalities using vector concatenation. The following section will describe how we obtain our visual and linguistic vectors as well as introduce the noun compound dataset that we use.

## 3   Data

To perform our experiments, we use two main sources of data: a relationally-labeled NNC interpretation dataset for training and testing Tratz (2011), and ImageNet (Deng et al., 2009) to extract visual feature embeddings. The following subsections will describe these datasets in more detail.

| | Split | Train | Val | Test |
|---|---|---|---|---|
| *Coarse* | random | 13835 | 928 | 3701 |
| | lexical (full) | 4650 | 1593 | 766 |
| | lexical (mod) | 9555 | 5316 | 3593 |
| | lexical (head) | 9048 | 5516 | 3900 |
| *Fine* | random | 13968 | 934 | 3725 |
| | lexical (full) | 4614 | 1574 | 843 |
| | lexical (mod) | 9511 | 5270 | 3846 |
| | lexical (head) | 8938 | 5640 | 4049 |

Table 1: Number of samples in each configuration (split and grain) of the Tratz (2011) dataset after our filtering.

## 3.1 Compound Dataset

Our main compound dataset for this work is a revised version of the Tratz (2011) noun compound dataset, which contains 19,158 distinct NNCs labeled with 37 fine-grained and 12 coarse-grained relation labels. The dataset is based on a previous one first published by Tratz and Hovy (2010), which contained 17509 compounds categorized by 43 fine-grained constituent relation labels. The compounds were annotated using Amazon's Mechanical Turk service.[2] They used a weighted majority-vote scheme based on ten annotation votes per compound, where Turkers voted on the quality on the other Turkers' decisions in order to even out potential inter-annotator disagreement. On their 43-class annotation task, Tratz and Hovy (2010) report a Cohen's $k$ (Cohen, 1960) of 0.57 as a measure of inter-annotator agreement.

To be able to test how compound interpretation models perform when dealing with unseen constituents, the Tratz (2011) dataset is split in various ways to ensure no previously seen constituents are available in the validation and testing phase. Different *lexical* splits ensure that the test and validation dataset contain no constituents previously seen in the training data – the *lexical mod* split ensures no previously seen modifiers (e.g. 'beach' in 'beach house'), the *lexical head* split ensures no previously seen heads (e.g. 'house'), and the *lexical full* split ensures no previously seen constituents at all. The *random* split does not take into account whether constituents are found in the training data or not.

Before performing our experiments, we do some filtering on the data in which we remove the fine-grained classes PERSONAL_NOUN, PERSONAL_TITLE, and LEXICALIZED. Our reason for removing the PERSONAL_NOUN and PER-

SONAL_TITLE classes is that there is some doubt as to whether proper names and titles possess the same semantic characteristics as common nouns (Cumming, 2007). Several works on NNC interpretation remove proper nouns from their data (e.g. Kim and Baldwin, 2006; Shwartz, 2019). Others, like (Dima and Hinrichs, 2015), choose to keep these categories but still acknowledge that their presence in the dataset is questionable. We remove the LEXICALIZED class because our work is mainly centered around how to interpret compounds that have a certain degree of compositionality, seeing as novel compounds, which likely make up the majority of compound types in most corpora, will need to be interpreted compositionally. Table 1 gives an overview of the number of samples in the train, validation, and test sets for each configuration of the Tratz (2011) dataset.

## 3.2 Image Data

ImageNet (Deng et al., 2009) is a large-scale image database which is structured using the WordNet (Miller, 1998) taxonomy, using synsets to represent sets of word meanings. Since many word classes are difficult to represent visually, ImageNet only contains nouns, and no other lexical categories, from the WordNet hierarchy. ImageNet contains 14,197,122 images, indexed by 21841 synsets[3], which represent different senses of the words.

**Selecting Synsets and Images from ImageNet** In order to collect the images needed for our task, we have to select all the synsets that were linked to each individual word in our dataset, and then retrieve the image URLs linked to those specific synsets. ImageNet is structured in such a way that one word can be linked to several synsets, and one synset can be linked to several words. Image URLs are associated with specific synsets, not specific words, so to retrieve an image URL from a word, one needs to first select which synset(s) one wants to use to represent that word.

Determining the appropriate sense to use for each constituent in a sample based on their context on the compound level is not trivial. We decide to go for a simple heuristic approach, namely finding the synset that most probably represents the most common or basic meaning for each word, given that the synset has images linked to it (where possible). Our heuristic method consists of the following steps:

[2] https://www.mturk.com/

[3] As per January of 2022

1. For a given word, let us call this our *target word*, retrieve all synsets that have images linked to them.

2. For each of the retrieved synsets, get the list of words that contain that synset among its synsets (representing the potential senses of the word). Let us call this list of words *comparison words*.

3. For each list of *comparison words*, compute the cosine similarity (by a pre-trained word2vec model) between each *comparison word* and the *target word*, and then take the average of all of these cosine similarities.

4. The synset whose *comparison words* list has the highest cosine similarity to the *target word* is selected as the most common, or basic, meaning.

Note that this method does not necessarily yield the *most common* sense, but the most common *imageable* sense, that is, the most common sense of a word, out of those which have related images. This choice was made on the basis of two assumptions: 1) it would give us the chance to collect more images, as opposed to selecting images only when the most common meaning is imageable,[4] and 2) an imageable synset that does not reflect the most common meaning of a word might still have certain visual properties in common with another less imageable, but more common, meaning of said word.

| | ImageNet | ResNet10 | ResNet100 | Total in data |
|---|---|---|---|---|
| Unique mods | 38.7% | 36.4% | 32.4% | 3126 |
| Unique heads | 40.1% | 37.6% | 31.6% | 3187 |

Table 2: Overview of the percentages of unique modifiers and heads in the coarse-grained random split of the (Tratz, 2011) data that have ImageNet images available, and that we could obtain ResNet$_{10}$ and ResNet$_{100}$ vectors for.

Table 2 gives an example of the ImageNet coverage of unique heads and modifiers in one dataset configuration (the random + coarse setting).

# 4 Methods

We frame the NNC interpretation task as a classification problem, experimenting with passing linguistic and visuo-linguistic vectors as inputs to an

SVM classifier. Our experimental process can be described in three steps:

1. Obtain linguistic vectors (from a pre-trained *word2vec* model) and visual feature vectors (from a pre-trained ResNet model) for the constituents of a compound (head and modifier). We experiment with both unimodal (word) embeddings, and visuo-linguistic embeddings, formed by concatenating the word embedding of a compound to the visual representation of a compound.

2. Combine the vector representations of each constituent (either linguistic, or visuo-linguistic).[5] We use two methods of combination: (a) simple concatenation, and (b) the Full Additive (FullAdd) method proposed by Zanzotto et al. (2010).

3. Observe and evaluate the performance of a setup depending on (a) modality of vectors (purely linguistic, or visuo-linguistic) and (b) mode of constituent vector combination.

To obtain linguistic vectors and visual vectors, we utilize pre-trained word2vec (Mikolov et al., 2013a) and ResNet (He et al., 2016) models, respectively. Our models, as well as our experimental setups and baselines, will be described in this section.

## 4.1 Models of Word Representation

We utilize a word2vec model (Mikolov et al., 2013a) to represent words in the linguistic modality, and visual vectors obtained by using a ResNet model (He et al., 2016) on ImageNet (Deng et al., 2009) images. The following subsections will describe these approaches in more detail.

### 4.1.1 word2vec

To obtain word embeddings to use as our linguistic vectors, we use a pre-trained word2vec model (Mikolov et al., 2013a). We employ a popular set of pre-trained word2vec embeddings that were trained on about 100 billion words from the GoogleNews dataset. These 300-dimensional word embeddings[6] were trained using a skip-gram approach with negative sampling (SGNS for short), as described in Mikolov et al. (2013b). Unlike previous work on

---

[4]In this case, we use 'imageable' to mean that ImageNet has images for it.

[5]In case a constituent lacks a vector representation in either modality, we instead use a vector of zeros.

[6]https://code.google.com/archive/p/word2vec/

this dataset published by e.g. Shwartz (2019), we decide to not train our own word2vec embeddings. This decision was made because our goal is investigating the effect of combining linguistic representations with visual ones, rather than comparing different kinds of linguistic representations, like Shwartz (2019) did.

### 4.1.2 ResNet

ResNet (He et al., 2016) is a deep residual neural network architecture for image recognition. ResNet models learn residual functions instead of unreferenced functions, allowing for the creation of models that are deeper than previous CNN models such as the VGG models (Chatfield et al., 2014), while still being less complex and faster to train (He et al., 2016). To extract visual embeddings based on images from ImageNet, we use a ResNet152 model trained on ImageNet data, implemented in the Keras (Chollet et al., 2015) library for Python. ResNet is trained on an object classification task, using 1.28 million images in its training phase. The model learns to take an image vector as input and outputs one out of the 1000 ImageNet category labels included in its training data.

To extract visual features using ResNet152, we flatten the final layer before the final classification (softmax) layer of the model, which has the size 7 x 7 x 2048, resulting in vectors of size 100352. Since a single, randomly selected image would not reflect all the potential visual aspects of an object, and finding the image that is closest to a prototypical representation of a concept is not trivial, we take the average of several image vectors to get a general visual representation for each noun. We use two experimental settings for visual features, where we extract and average feature vectors for 10 or 100 ImageNet images. We will refer to these vectors as $ResNet_{10}$ and $ResNet_{100}$, respectively. See Table 2 for a summary of the image availability in the Tratz (2011) dataset. These vectors can then be reduced to our desired vector dimensions, for example 300 in order for them to be compatible with pre-trained 300-dimensional *word2vec* embeddings. For our ResNet vectors to be more appropriate as inputs to our SVM classifiers, we scale our vectors so that the values range from -1 to 1.

### 4.2 Modes of Vector Combination

To combine modifier and constituent vectors into compound vectors, we test two different modes of combination: simple vector concatenation, and the FullAdd model (Zanzotto et al., 2010). In both cases, the combination of a modifier vector and a head vector only happens within one modality, i.e. we would not combine a linguistic modifier vector with a linguistic head vector. For our visuo-linguistic setups, compound vectors are composed in each modality and then the resulting vectors are concatenated to form a visuo-linguistic representation of the compound. The following subsection will describe the FullAdd model in more detail.

### 4.2.1 The Full Additive Model

The Full Additive model, also referred to as FullAdd or the Estimated Additive model (Zanzotto et al., 2010) is a model where the two vectors $\overrightarrow{x}$ and $\overrightarrow{y}$, representing the constituent words $c_1$ and $c_2$, are multiplied by square matrices $A$ and $B$, respectively, and then added together to create a compositional meaning representation of a phrase. $A$ and $B$ are the same for each vector $\overrightarrow{x}$ and $\overrightarrow{y}$, respectively, and are obtained through training on a training set of compound nouns that contains distributional vector representations of each compound and each constituent word. We can think of these vectors as being ordered in triples, where any triple of words *(z, x, y)*, which corresponds to *(compound, modifier, head)* in English, is represented by a triple of vectors $(\overrightarrow{z}, \overrightarrow{x}, \overrightarrow{y})$. For example, the training set could contain the vector triple $(\overrightarrow{soap\ opera}, \overrightarrow{soap}, \overrightarrow{opera})$. The goal will be to learn a composition function for any word vectors $\overrightarrow{x}, \overrightarrow{y}$ such that $\overrightarrow{p} = f(\overrightarrow{x}, \overrightarrow{y})$ approximates $\overrightarrow{z}$, where $\overrightarrow{p}$ is the composed vector for any given noun compound, and $\overrightarrow{z}$ is the observed distributional vector for this noun compound. In other words, the function is trained using compounds for which we have a distributional representation, and can then be used to create compositional representations of compounds where a distributional representation is not available.

Intuitively, one can think of the process of training the two matrices (one for modifiers and one for heads) as finding a way of transforming a meaning representation of a single word into its *as-constituent* meaning. For example, by multiplying the vector for *chocolate* with the modifier matrix (which we call matrix *A*), we approximate the *as-modifer* meaning of *chocolate*, as in *chocolate cake*. The general equation for composing a compound vector $\overrightarrow{z}$ given two constituent word vectors $\overrightarrow{x}$ and $\overrightarrow{y}$ is given below:

$$\vec{z} = \mathbf{A}\,\vec{x} + \mathbf{B}\,\vec{y} \qquad (1)$$

To implement our FullAdd model, we use the Distributional Semantic Composition Toolkit, or DISSECT (Dinu et al., 2013), which allows for the implementation of FullAdd as well as other composition models. To prepare the necessary data for FullAdd, we filter our training data so that we only keep the compounds for which the whole compound as well as the modifier and head separately have vectors associated with them in our word embedding model. Then we construct a semantic space using those word embeddings. Due to the requirements of the DISSECT implementation, heads and modifiers cannot be repeated in the space (e.g., we can not include both 'cat food' and 'dog food'). The two FullAdd matrices, $\mathbf{A}$ and $\mathbf{B}$, can then be trained in the way described above. The resulting vectors are then used to compose compositional meaning vectors for our training, test, and validation data. In our FullAdd experiments, we train a FullAdd model for each modality (linguistic and visual) and then create composed vectors for each compound in each modality before combining the two modalities using concatenation.

### 4.3 Experimental Setups

For our experiments, we create three majority-class baselines in addition to our SVM classifier.[7] In this section, we will describe our baselines and our main experimental setups.

#### 4.3.1 Baselines

We implement the following majority-class baselines:

- Overall Majority: For a given data sample, this baseline selects the overall majority class as observed in the training data.

- Modifier Majority: For a given data sample, this baseline selects the majority class represented among compounds in the training data with the same modifier as the sample.

- Head Majority: For a given data sample, this baseline selects the majority class represented among compounds in the training data with the same head as the sample.

---

[7] We did also perform a few NNC interpretation experiments using a BERT model, which were not included in this paper because of poor performance on the lexical splits of the Tratz (2011) dataset. See Table 7 in the appendix for an overview.

The 'Modifier Majority' and 'Head Majority' rely on using the modifiers or heads, respectively, from the training data to determine the assigned label of each data sample. However, we have several dataset configurations in which the training and test datasets do not share any heads, modifiers, or any constituents at all – see Table 1 for a summary. Thus, in these configurations, the head or modifier majority mechanism will not work. This means that for the lexical + mod split of our data, the Modifier Majority baseline will give the exact same results as the Overall Majority baseline. The same is the case for the lexical + head split together with the Head Majority baseline, as well as the full lexical split with both the Modifier Majority and Head Majority baselines.

#### 4.3.2 Classifier Setup

Our classifier is an SVM that takes as inputs either linguistic representations (in the form of *word2vec* vectors that have either been concatenated or composed using the FullAdd function) or visuo-linguistic representations (in the form of linguistic vectors concatenated with the visual vectors described in section 4.1.2). We use an SVM with a one-vs-rest scheme for multiclass classification. The SVM has a linear kernel, L2 penalty and a C value of 0.5. We train our classifier on the Tratz (2011) data, passing either our linguistic or visuo-linguistic vectors as inputs.

## 5 Results and Evaluation

We evaluate our setup on the Tratz (2011) dataset and report F1 scores for all dataset configurations.

| | Split | MC-O | MC-M | MC-H |
|---|---|---|---|---|
| *Coarse* | random | 7.5 | 40.0 | 59.3 |
| | lexical (full) | 6.7 | – | – |
| | lexical (mod) | 7.8 | – | 58.8 |
| | lexical (head) | 7.0 | 38.6 | – |
| *Fine* | random | 5.3 | 34.5 | 54.1 |
| | lexical (full) | 5.6 | – | – |
| | lexical (mod) | 6.3 | – | 52.6 |
| | lexical (head) | 5.2 | 34.8 | – |

Table 3: F1 scores from our baseline classifiers. MC stands for Majority Class; O stands for Overall, M for Modifier, and H for Head.

Table 3 shows the weighted F1 scores of our baseline classifiers. The modifier- and head-majority classifiers require the test datasets to include previously seen modifiers and heads (respectively), which is why the table has some cells that

are marked with '–', indicating that the score for this cannot be computed with the given majority-class strategy and thus would get the same score as the overall majority baseline. For this reason, we only have comparable scores from the modifier- and head-majority classifiers in the case of the random split, in which both the fine-grained setting and the coarse-grained setting show that the head-majority classifier performs the best. In other words, it seems that having a common head is a greater indicator of same-class membership than having a common modifier in the Tratz (2011) dataset.

| | Split | w2v | w2v + ResNet10 | | w2v + ResNet100 | |
|---|---|---|---|---|---|---|
| | random | 66.3 | 66.0 | - 0.3 | **66.4** | + 0.1 |
| Coarse | lexical (full) | **44.2** | 44.1 | - 0.1 | 43.7 | - 0.5 |
| | lexical (mod) | 57.9 | **58.3** | + 0.4 | 57.7 | - 0.2 |
| | lexical (head) | 50.8 | 51.0 | + 0.2 | **51.3** | + 0.5 |
| | random | **66.7** | 66.6 | - 0.1 | **66.7** | +/- 0 |
| Fine | lexical (full) | 39.2 | **39.4** | + 0.2 | 38.4 | - 0.8 |
| | lexical (mod) | 56.4 | 56.4 | +/- 0 | **56.5** | + 0.1 |
| | lexical (head) | 47.1 | **47.5** | + 0.4 | 46.9 | - 0.2 |

(a) F1 scores using FullAdd-composed compound vectors

| | Split | w2v | w2v + ResNet10 | | w2v + ResNet100 | |
|---|---|---|---|---|---|---|
| | random | 74.1 | **75.3** | + 1.2* | 75.2 | + 1.1* |
| Coarse | lexical (full) | 49.1 | **50.8** | + 1.7* | 50.0 | + 0.9 |
| | lexical (mod) | 63.5 | **64.0** | + 0.5 | 63.1 | - 0.4 |
| | lexical (head) | 55.5 | **56.7** | + 1.2 | 56.0 | + 0.5 |
| | random | 73.0 | **75.0** | + 2.0 | **75.0** | + 2.0 |
| Fine | lexical (full) | 40.3 | 40.0 | - 0.3 | **40.7** | + 0.4 |
| | lexical (mod) | 63.0 | **63.5** | + 0.5 | 63.4 | + 0.4 |
| | lexical (head) | 50.6 | 51.6 | + 1.0* | **52.0** | + 1.4* |

(b) Results using concatenated compound vectors

Table 4: Weighted F1 scores from classification experiments using linguistic and visuo-linguistic vectors. The tables show results of using FullAdd-composed vectors as well as concatenation-composed vectors, with the change in F1 obtained when ResNet vectors are included. An asterisk next to an increase in F1 score means the bimodal result is significantly different from its unimodal counterpart following a Bonferroni-corrected McNemar test.

Table 4 shows the results of our experiments on the Tratz (2011) data after our filtering. All of the scores given in the tables are F1 scores, and an asterisk next to an increase in score means that the increase was found to be significant following a McNemar test (McNemar, 1947) and a Bonferroni correction (Neyman and Pearson, 1928) of the p-values.[8] As is evident when comparing tables 4a and 4b, using concatenated vectors as opposed to FullAdd composed vectors yields much higher F1 scores. Additionally, the results in table 4b,

---

[8]We set our $\alpha$ level to the conventional 0.05, which resulted in a p-value threshold of 0.00625 after a Bonferroni correction.

with concatenated vectors, are less ambiguous: in this experiment, at least one of the visuo-linguistic settings beats the purely linguistic setting in each experimental setting.

As has been shown in previous research on this dataset, the most challenging dataset split is the full lexical split, where no constituents in the validation and test data are previously seen in the training data. As expected, the fine-grained setting is generally more challenging than the coarse-grained one. As we can see from comparing tables 4a and 4b, the results in the former table are more ambiguous, meaning that we cannot conclude that one input type (linguistic or visuo-linguistic vectors) is better than another. In table 4b, however, we find that our visuo-linguistic vectors help increase scores in some cases. In the case of the ResNet$_{10}$ vectors, the increase in scores is significant for the random and full lexical splits in the coarse-grained setting as well as for the lexical (head) split in the fine-grained setting. For our ResNet$_{100}$ vectors, only the coarse + random and the fine + lexical (head) settings show a significant increase in scores. We find small increases overall for most NNC relation classes, rather than big increases for certain relation classes (see Figure 3 in Appendix A for an example).

Table 5 shows results of experiments run on a subset of the data for which ResNet$_{10}$ vectors were available for both the modifier and head of each compound. We compare results on our baselines as well as our FullAdd and concatenation models with textual or visual vectors alone, on the same subset. As with the results on the full dataset, the concatenation method performs better than the Full-Add model here. Additionally, it seems that the visual vectors do contribute at least some valuable information on their own. It is important to note that Table 5 is not directly comparable to the tables in 4, since the former shows results on just a small subset of the data.

One might be inclined to question why our Full-Add experiments on this dataset perform worse than very similar experiments done by e.g. Shwartz (2019) and Dima (2016). This is likely due to the fact that Shwartz (2019) and Dima (2016) trained their own word embeddings specifically for this task, meaning that they were able to obtain distributed embeddings for more of the compounds in the Tratz (2011) dataset than what we had available through our pre-trained model, and as a con-

| grain | split | baselines | | | word2vec | | ResNet10 | |
|---|---|---|---|---|---|---|---|---|
| | | MC-O | MC-M | MC-H | FullAdd | concatenate | FullAdd | concatenate |
| *Coarse* | random | 9.8 | 40.6 | 48.4 | 56.0 | 70.7 | 30.9 | **62.7** |
| | lexical (full) | 3.9 | – | – | 26.2 | 41.9 | **6.9** | **28.5** |
| | lexical (mod) | 6.3 | – | 47.1 | 45.7 | 59.7 | 20.4 | 47.0 |
| | lexical (head) | 5.7 | 32.0 | – | 30.8 | 47.1 | 19.8 | **40.4** |
| *Fine* | random | 3.3 | 32.5 | 38.9 | 53.7 | 66.6 | 16.9 | **59.1** |
| | lexical (full) | 4.6 | – | – | 23.6 | 31.2 | **5.7** | **18.6** |
| | lexical (mod) | 2.8 | – | 36.1 | 34.4 | 51.3 | 10.7 | **41.1** |
| | lexical (head) | 3.4 | 33.8 | – | 36.5 | 41.7 | 10.8 | **34.4** |

Table 5: Results (reported in F1) from experiments with unimodal vectors in either modality (word2vec vectors alone or ResNet$_{10}$ vectors alone) on a subset of the data for which ResNet$_{10}$ vectors were available. Baseline results on the same subset are included for comparison. Scores in bold are cases in which the ResNet$_{10}$ vectors outperform the strongest baseline.

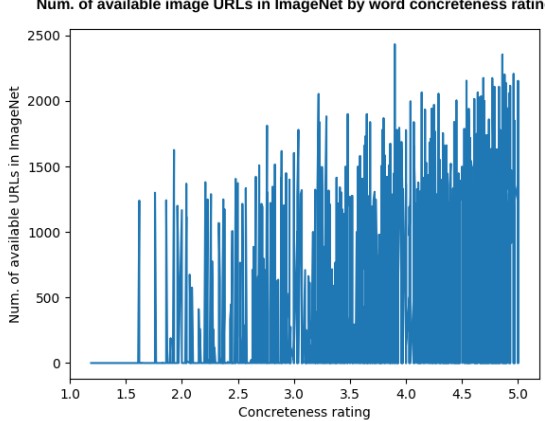

Figure 2: Word concreteness rating by number of available URLs in ImageNet

sequence had more training data for the FullAdd model. As our goal with this work is not to compare composition functions for linguistic vectors, we saw training our own embeddings as being superfluous for this study.

Overall, we see that, in our experiments with concatenated compound vectors, adding visual information helps increase the scores in all cases, and in some cases the increases are statistically significant.

## 5.1 Concreteness Ratings

Intuitively, one could assume that visual information (i.e. images) would be easier to obtain for more concrete words, thus making visual information a more appropriate and/or helpful addition for compounds that have relatively concrete constituents. If this is the case, then we should find a higher benefit of incorporating visual information, the more concrete a word is.

We quantify concreteness using a dataset of concreteness ratings of almost 40,000 English lemmas, by Brysbaert et al. (2014). The ratings are continuous values between 1 and 5, where 5 is the most concrete. The ratings were obtained by surveying more than 4,000 participants in a crowdsourcing study and taking the mean of the ratings obtained for each word.

As a first analysis of our results in light of these concreteness ratings, we performed several logistic regression analyses where we looked at the concreteness ratings of modifiers and heads as predictors of classification success. Table 6 in Appendix A gives a full overview of these results. What we find is that the dataset configuration seems to matter more than the modality, but that the concreteness ratings of both modifiers and heads are, in some cases, good predictors of classifier success. However, in the significant cases, we discover a negative

relationship between concreteness and classification success – i.e., the higher the concreteness of a modifier/head, the lower the chance of the classifier predicting the correct class. We performed an investigation into some of our results, filtering the samples by image availability (specifically, whether a constituent had fewer or more than 10 images available in ImageNet). The full results are found in Table 8 in Appendix A.

Figure 2 shows word concreteness ratings by number of URLs available in ImageNet, as determined by our image selection heuristic, for each of the words in the Tratz (2011) dataset that had concreteness ratings in the Brysbaert et al. (2014) dataset (regardless of whether they appeared as a modifier or head).

A correlation analysis revealed a low to moderate correlation between the concreteness ratings and the URL counts (Pearson's r = 0.45, p < 0.001). This indicates that, to some extent, the higher the concreteness rating of a constituent in a compound, the higher the chances of finding 10 or 100 images to represent said constituent as part of our image vectors. Yet, in experiments on the subset of compounds for which both constituents had ResNet$_{10}$ vectors available, we find that our visual vectors alone are somewhat informative, as we saw in Table 5. Examples of words for which we were not able to obtain at least 10 images include *minute* (concreteness rating 3.04), *intelligence* (concreteness rating 2.24), and *state* (concreteness rating 3.52).

The negative relationship between constituent concreteness and classifier success seems counterintuitive, but might be a result of a number of fac-

tors related to word frequency, polysemy, and the distribution of concrete vs. non-concrete words over the classes in the Tratz (2011) dataset. Although one might expect compounds containing concrete constituents to benefit more from visuo-linguistic representations, we note that the negative correlation between concreteness and classification success is always found in the visuo-linguistic modality whenever it is found in the linguistic modality. In other words, this seems to be a general finding rather than a modality-specific one. As suggested by previous work, concrete and abstract words differ in the kinds of contexts they tend to appear in, where abstract words tend to occur near other abstract words, and concrete words occur in more varied contexts (Frassinelli et al., 2017). Additionally, it has been found that distributional semantic models like word2vec are worse at modeling word pair similarity for highly concrete words than for highly abstract words (Hill et al., 2015). Since our task is relation classification, our findings might also be partially influenced by the distribution of relation labels for concrete and non-concrete words. For example, abstract words may be more restricted in which relations they can partake in, and thus be easier to classify. We leave it up to future work to investigate these relationships, but we note that our visuo-linguistic representations do tend to outperform the purely linguistic ones, regardless of constituent concreteness ratings.

## 6   Conclusion and Future Work

In this paper, we have presented NNC interpretation experiments on the Tratz (2011) dataset, comparing linguistic and visuo-linguistic inputs to an SVM classifier. We have found that, in our best-performing case, concatenating visual feature vectors with linguistic vectors (word embeddings) helps increase F1 scores on the Tratz (2011) dataset in almost all experimental settings. Our findings indicate that utilizing visual information for this NNC relation classification task might indeed be a promising endeavor.

Future work should aim to further refine our approach by for example using more sophisticated methods for selecting images to represent words, exploring ways to represent abstract or non-imageable words, and finding better ways to visually ground polysemous words. In this regard, recent multimodal encoders pretrained on visual and linguistic data (e.g. Lu et al., 2019; Tan and

Bansal, 2019), are a promising way forward. Another possible angle for future work could be to consider NNC interpretation in visual and linguistic contexts. In the future, we would also be eager to explore visual grounding in other aspects of computational NNC related tasks, such as NNC generation. Additionally, our approach should be tested on different datasets and in different circumstances, for example in a task that determines the probability of compound categories rather than fixed classes. One final potential angle for future work could be to look further into the task of visual composition. A first step could be to more closely examine the effects of using the FullAdd function with image vectors.

To conclude, our results are in line with previous work from both cognitive science and computational linguistics suggesting that more psychologically plausible models of NNC processing should incorporate grounding.

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

# Appendix

## A Detailed Results Tables

This appendix contains supplementary tables that describe some of our findings in more detail.

Figure 3 shows the F1 scores for each relation in the random + coarse dataset configuration for word2vec vectors and word2vec + $ResNet_{10}$ vectors. As opposed to Table 5, Figure 3 shows results from our full dataset, rather than the subset of compounds with 'imageable' constituents.

| | | modifiers | | | | heads | | | |
| | | L | | VL | | L | | VL | |
| split | | coef | p | coef | p | coef | p | coef | p |
|---|---|---|---|---|---|---|---|---|---|
| Coarse | random | **-0.2049** | **<0.001** | **-0.1675** | **<0.001** | -0.18 | 0.008 | -0.99 | 0.028 |
| | lexical (full) | -0.1788 | 0.032 | -0.1251 | 0.133 | 0.076 | 0.352 | 0.13 | 0.113 |
| | lexical (mod) | **-0.2322** | **<0.001** | **-0.2740** | **<0.001** | **-0.1924** | **<0.001** | **-0.2093** | **<0.001** |
| | lexical (head) | **-0.1710** | **<0.001** | **-0.2118** | **<0.001** | 0.0053 | 0.888 | 0.0432 | 0.252 |
| Fine | random | **-0.2128** | **<0.001** | **-0.2073** | **<0.001** | **-0.268** | **<0.001** | **-0.27** | **<0.001** |
| | lexical (full) | 0.1108 | 0.171 | 0.1665 | 0.041 | 0.0854 | 0.340 | 0.0296 | 0.741 |
| | lexical (mod) | -0.1340 | 0.001 | **-0.1581** | **<0.001** | **-0.3257** | **<0.001** | **-0.3306** | **<0.001** |
| | lexical (head) | 0.0393 | 0.278 | 0.0423 | 0.243 | 0.0659 | 0.090 | 0.0569 | 0.144 |

Table 6: Results from a logistic regression analysis of modifier and head concreteness as a predictor of the successful classification of compounds. The scores in boldface are ones where the p-values are lower than a Bonferroni-corrected $\alpha$ level of 0.05.

Table 6 contains a summary of several logistic regression analyses performed on our classification results in both the linguistic and visuo-linguistic modalities. The results show coefficients and p-values of analyses using modifier and head concreteness (separately) as predictors of classification success.

| split | grain | BERT | BERT + $ResNet_{10\_RAW}$ | | | BERT + $ResNet_{10\_NORM}$ | | |
|---|---|---|---|---|---|---|---|---|
| | | | F1 | diff | $\epsilon, (B<BM_{RAW})$ | F1 | diff | $\epsilon, (B<BM_{NORM})$ |
| random | coarse | **78.7** | 69.7 | - 9 | 0.99 | 78.7 | +/- 0 | 0.47* |
| random | fine | **57.9** | 50.7 | - 7.2 | 0.94 | **65.1** | + 7.2 | 0.024* |
| lexical (full) | coarse | **31.6** | 28.6 | - 3 | 0.92 | 25.8 | - 5.8 | 0.95 |
| lexical (full) | fine | **19.5** | 14.5 | - 5 | 1.00 | 15.0 | - 4.5 | 0.79 |
| lexical (mod) | coarse | 17.0 | **36.6** | + 19.6 | 0.036* | 6.7 | - 10.3 | 0.98 |
| lexical (mod) | fine | 8.3 | **27.0** | + 18.7 | 0.005* | 8.3 | +/- 0 | 0.55 |
| lexical (head) | coarse | 11.1 | **40.2** | + 29.1 | 0.004* | 5.0 | - 6.1 | 0.84 |
| lexical (head) | fine | 4.4 | **29.5** | + 25.1 | 0.036* | 2.8 | - 1.6 | 0.78 |

Table 7: Results from fine-tuning BERT with and without adding $ResNet_{10}$ vectors after 50 epochs of training, averaged over 10 runs. Each column of bimodal results shows weighted F1, the change in F1 between the unimodal and the given bimodal setting, and the epsilon value from the ASD algorithm that reveals to what extent the bimodal is better than the unimodal setting.

Table 7 shows the results of some NNC interpretation experiments that we did with a pre-trained BERT model (Devlin et al., 2018) and our ResNet visual embeddings. In these experiments, we fed compounds to a BERT model fitted with a linear classifier on top in order to get the classifications of the compounds. In the visuo-linguistic modality, we concatenated BERT's linguistic embeddings with our visual embeddings before passing them to a linear classification layer. We experimented with using raw ResNet embeddings (straight out of the ResNet model, without applying anything but dimensionality reduction) and normalized ResNet embeddings. The table shows F1 scores as well as the $\epsilon$ value returned by the *Almost Stochastic Dominance* (ASD) algorithm proposed by Dror et al. (2019) for comparing the performance of two neural network architectures. The algorithm works in such a way that an $\epsilon$ value of less than 0.5 means that algorithm B (in our case, one of the visuo-linguistic settings) is *almost stochastically dominant* over algorithm A (in our case, the purely linguistic setting).

Table 8 gives an overview of the results of our classification algorithm when used on linguistic (L) and visuo-linguistic (VL) vectors. The table shows the F1 scores for subsets of our test data, where we select data samples where either one, both, or none of the constituents in each sample had a $ResNet_{10}$ vector available (i.e., had 10 or more images available in ImageNet). The 'no filtering' column contains the exact same results, for the full dataset, as reported in our main article, and is included for comparison.

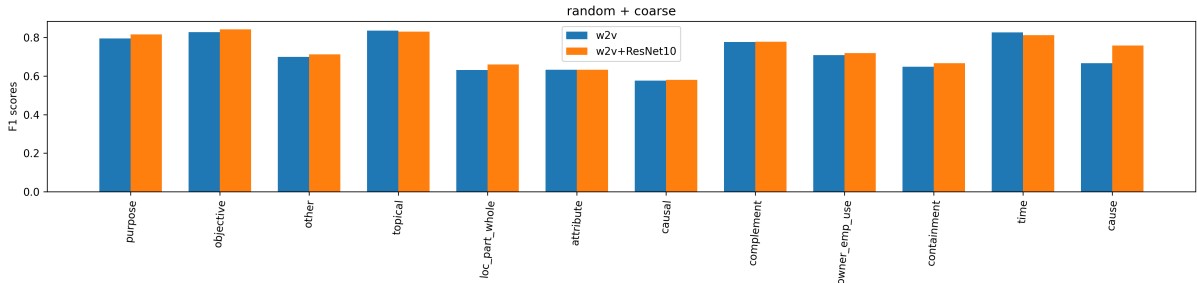

Figure 3: Per-relation F1 scores in the condition with the highest scores (the random + coarse configuration).

| constituents with 10+ images: | | no filtering | | mods | | heads | | both | | none | |
|---|---|---|---|---|---|---|---|---|---|---|---|
| **split** | | **L** | **VL** | **L** | **VL** | **L** | **VL** | **L** | **VL** | **L** | **VL** |
| *Coarse* | random | 74.1 | **75.3*** | 70.46 | 71.43 | 72.78 | 74.65 | 70.76 | 72.59 | 76.84 | 77.9 |
| | lexical (full) | 49.1 | **50.8*** | 41.29 | 46.04 | 48.4 | 51.24 | 41.3 | 43.22 | 56.84 | 53.77 |
| | lexical (mod) | 63.5 | 64.0 | 56.77 | 57.25 | 59.36 | 61.35 | 54.48 | 55.45 | 70.22 | 69.69 |
| | lexical (head) | 55.5 | 56.7 | 54.3 | 55.7 | 50.85 | **54.64*** | 51.42 | **55.96*** | 58.99 | 59.14 |
| *Fine* | random | 73.0 | 75.0 | 69.49 | **71.72*** | 69.19 | **72.06*** | 66.13 | 69.43 | 76.09 | **77.78*** |
| | lexical (full) | 40.3 | 40.0 | 41.19 | 40.91 | 39.21 | 35.04 | 42.03 | 38.94 | 42.33 | 44.59 |
| | lexical (mod) | 63.0 | 63.5 | 60.13 | 60.2 | 54.32 | 55.77 | 51.88 | 53.38 | 68.71 | 69.35 |
| | lexical (head) | 50.6 | **51.6*** | 52.06 | 53.35 | 51.18 | 51.6 | 52.88 | 53.91 | 50.17 | 51.53 |

Table 8: Results of our experiments using the concatenation method of composition and the ResNet$_{10}$ vectors, filtered by the imageability (as modeled by whether or not 10 or more images were available) of the constituents. An asterisk next to a VL score means that the visuo-linguistic (VL) modality performed significantly better than the linguistic (L) modality following a McNemar test with a Bonferroni correction of the p-values.