# OpenReview forum: "Visually Grounded Interpretation of Noun-Noun Compounds in English"
_aclweb.org/ACL/2022/Workshop/CMCL — CMCL 2022_

### Official Review · Reviewer_JpgF · 2022-03-23
**Seems to work, but the prose oversells the results**

**Rating:** 6
**Confidence:** 4

**Review:**

This paper investigates whether and to what extent augmenting linguistic representations with visual representations can improve a system's classification of noun-noun compounds (NNCs). NNCs come from the Tratz (2011) data set. They try a few different ways of combining the linguistic and visual representations.

I appreciated that they specified several data splits up front: course vs fine-grained labels, and train/dev/test splits controlling for both the modifier and head members of the NNCs. This allows, in theory, for a lot of interesting comparison and analysis.

One major challenge which the authors faced was how to match up images from ImageNet with the Tratz NNCs. They present an algorithm to do this which seems reasonable, but is still quite approximate. This bring this up again in the conclusion. My worry is that their results might be greatly affected by their matching heuristic rather than the problem at hand, and while this is a great topic for a follow-up, I wish they had addressed it in this paper. I believe it is more critical than they made it out to be.

They present F1 scores for a family of 3 baseline models which take majority classes of the entire compound, head, or modifier in the training data. Providing a sensible baseline is something that is often skipped these days, so I appreciated this.

They test a concatenate and an additive model for combining ResNet visual representations with word2vec linguistic representations.

They find similar but not identical results for the additive and concatenative model. The latter performed slightly better on average, and in both cases they generally outperform word2vec alone. I was a bit surprised when I got to the results though, because +ResNet really is only a little bit better than word2vec alone. It seemed kind of incongruous with the big claims made in the prose of the paper. It's sort of trivial to assume that including additional informative data (as through ResNet) should help performance if it's combined correctly, and that is what we see from the results. So my question is, are these results dramatic enough to be interesting? And why aren't they better? Was it because of the matching heuristic? I don't know

Also not sure what part of this paper is meant to be cognitive modeling.

---

### Official Review · Reviewer_rwmZ · 2022-03-24
**A useful contribution in need of some clarity/reframing**

**Rating:** 6
**Confidence:** 3

**Review:**

This is a well-written paper; the authors clearly lay out the objective, the previous work on the topic, and their contribution. However, the reported performance increases are very modest, and some methodological choices are unclear. Overall, I think this paper is appropriate for CMCL if the authors adjust the framing and provide some clarifications.

1.	The main claim is that concatenating word2vec and visual (ResNet) activations will improve interpretation (classification) of compound nouns by providing additional semantic information. However, the actual increases are very small (an average of 1% or less). It would be helpful if the authors acknowledged that the increases are actually quite modest. Moreover, Table 5 in the appendix shows that comparable performance gains can be observed even in cases where the nouns do not have a visual representation (so their final representation is word2vec + zeros). Thus, the utility of visual representations remains debatable.
2.	ImageNet representations are obtained by averaging representations of several images. Has that been shown to be a useful approach for representing visual concepts? I would imagine that even if providing visual information is beneficial, averaging several distinct image vectors might make the visual representation less useful than it could have been if, say, the most prototypical image was chosen (this might depend on the specific object type too, e.g. less concrete objects are likely more visually diverse).
3.	I don’t quite get the way the Full Additive Model was implemented. Were the x, y, and z vectors pretrained word2vec vectors? More generally, you state toward the beginning that the goal is to learn a way to represent compounds compositionally, yet if the compound is frequent enough to have its own ground truth representation (z), doesn’t it violate the compositionality premise?
4.	Is there a reason why visual reps alone aren’t used for classification? Seems like a useful baseline.
5.	The authors address the question of whether word concreteness affects their results by (a) examining how many imagenet images each word is associated with, (b) performing additional tests of whether image availability and concreteness affects classification accuracy. However, a key question here seems to be addressed only indirectly: do compounds with concrete words benefit more from adding visual representations? This could be done by dividing the compounds into 2 groups, concrete and abstract, and examining whether the benefits of adding visual reps vary between these two groups. Alternatively, the tests in Table 6 can be complemented by an explicit comparison of relative concreteness effects in L and VL modalities. The question of whether adding image vectors benefits only highly concrete words arose for me as soon as I read the title/abstract, so I think it’d be important to address it a bit more.

---

### Official Review · Reviewer_Zq1M · 2022-03-24
**Well written paper with unimpressive results**

**Rating:** 6
**Confidence:** 3

**Review:**

The authors describe a classification experiment in which they mixed linguistic and visual features to predict the semantic relation holding between the components of a noun noun compound.

By itself the paper is well written and the topic very interesting.

However, I find the reported results a bit unimpressive due to a series of reasons:
- first of all, the differences in the reported F-measures are mostly almost negiglible. The authors should at least comment on that
- the negative correlation between concreteness score and classification success is really puzzling, almost worrying. From an intuitive point of view, it suggest that the visual information (as encoded in this experiment) is not really that useful
- it is not clear why the authors didn't test the performance of the visual features in isolation
- a crucial piece of information that is missing from the paper is an analysis of the relation-wise perfomance of the best-performing classifier

---

### Decision · Program_Chairs · 2022-03-29

Accept